# Quantifying Abdominal Coloration of Worker Honey Bees

**DOI:** 10.3390/insects15040213

**Published:** 2024-03-22

**Authors:** Jernej Bubnič, Janez Prešern

**Affiliations:** 1Animal Production Department, Agricultural Institute of Slovenia, 1000 Ljubljana, Slovenia; janez.presern@kis.si; 2Biotechnical Faculty, University of Ljubljana, 1000 Ljubljana, Slovenia

**Keywords:** honey bees, temperature, phenotypic plasticity, image analysis, abdominal coloration, image classification

## Abstract

**Simple Summary:**

We obtained four frames of honey bee brood from two colonies and incubated them at two different temperatures (30 and 34 °C). One colony had workers exhibiting yellow marks on the abdomen, while the other did not. We collected hatched workers and photographed abdomens. Images were analyzed using custom-written R script to obtain vectors that describe the coloration on the length of abdomen and were summarized in a single value—coloration index. We used Uniform Manifold Approximation and Projection (UMAP) to reduce the dimensions of the vectors and to develop a classification procedure with the support vector machine method. We tested the effect of brood origin and temperature on coloration index with ANOVA. UMAP did not distinguish individual abdomens according to experimental group. A trained classifier sufficiently separated the abdomens incubated at different temperatures. We improved the performance by preprocessing data with UMAP. The differences in mean coloration index were not significant among the gray groups incubated at different temperatures nor between the yellow groups. However, the differences were significant between the gray and yellow groups. The developed color-recording protocol and statistical analysis provide useful tools for quantifying abdominal coloration in honey bees. The developed coloration index has the potential for determining subspecies of honey bees. Our results indicate that the environmental temperature in the selected range during development does not seem to impact honey bee coloration significantly.

**Abstract:**

The main drawback in using coloration to identify honey bee subspecies is the lack of knowledge regarding genetic background, subjectivity of coloration grading, and the effect of the environment. The aim of our study was to evaluate the effect of environmental temperature on the abdominal coloration of honey bee workers and to develop a tool for quantifying abdominal coloration. We obtained four frames of honey bee brood from two colonies and incubated them at two different temperatures (30 and 34 °C). One colony had workers exhibiting yellow marks on the abdomen, while the other did not. We collected hatched workers and photographed abdomens. Images were analyzed using custom-written R script to obtain vectors that summarize the coloration over the abdomen length in a single value—coloration index. We used UMAP to reduce the dimensions of the vectors and to develop a classification procedure with the support vector machine method. We tested the effect of brood origin and temperature on coloration index with ANOVA. UMAP did not distinguish individual abdomens according to experimental group. The trained classifier sufficiently separated abdomens incubated at different temperatures. We improved the performance by preprocessing data with UMAP. The differences among the mean coloration index values were not significant between the gray groups incubated at different temperatures nor between the yellow groups. However, the differences between the gray and yellow groups were significant, permitting options for application of our tool and the newly developed coloration index. Our results indicate that the environmental temperature in the selected range during development does not seem to impact honey bee coloration significantly. The developed color-recording protocol and statistical analysis provide useful tools for quantifying abdominal coloration in honey bees.

## 1. Introduction

Abdominal coloration of honey bees is arguably the earliest morphological characteristic used to describe the honey bee subspecies, at least by non-scientists [1,2]. The main disadvantage in using abdominal coloration as an indicator of the honey bee subspecies is the lack of knowledge regarding genetic background and the subjectivity of coloration grading. In addition, coloration can be affected by the environment compared to some other morphological parameters used for determining honey bee subspecies [3].

Abdominal coloration in honey bees is a trait with high heritability that varies by cast and gender. The heritability of color patterns is 0.39 in drones, 0.32 in workers, and 0.21–0.23 in queens [4]. Roberts and Mackensen [5] concluded that abdominal coloration is regulated by seven genes. Woyke [6] later concluded that abdominal coloration is regulated by only three major genes that are regulated by six to seven polygenes.

The environment to which the honey bees are exposed as pupae, especially the temperature, plays an important role in the abdominal coloration of worker bees [7,8]. A change in the abdominal coloration in workers of *Apis cerana* can be induced by manipulating the environmental temperature to which workers are exposed as pupae [8]. Workers that were exposed to higher temperatures as pupae later exhibited yellow coloration more frequently than those exposed to lower temperatures. Degrandi-Hoffman et al. [7] obtained similar results in their experiment with *A. mellifera* commercial queens. Their queens exposed to higher temperatures as pupae were lighter in color as adults than those exposed to lower temperatures. Gruber et al. [9] reported that the morphometric properties of honey bees in Ethiopia vary with altitude. At lower altitudes, honey bees express yellow phenotypes and at higher altitudes express darker phenotypes. Yellow and dark Ethiopian honey bees are considered as two subspecies (*Apis mellifera monticola* and *Apis mellifera scutellata*) even though they are genetically very similar [10]. A honey bee colony maintains a temperature between 33 and 36 °C, with the nest center being warmer than the edges [11]. Thus, the position of the brood in the nest could affect the body coloration and contribute to anecdotal changes in coloration over the years.

Despite the lacking understanding of biological mechanisms driving the coloration of the abdomen, this trait is actively used in many breeding programs. The dark coloration of the native honey bees is an important factor in the Irish and Slovenian breeding programs for *Apis mellifera mellifera* and *Apis mellifera carnica* [12,13], while queen breeders in the Azores prefer colonies that express yellow coloration [12]. The coloration of honey bee workers is also used for assessing the rate of hybridization between subspecies in some breeding programs. Coloration is assessed at the first, second, and third abdominal tergites. The degree of coloration is assessed using different scales [7,8,14,15] as described by Ruttner [2]. Coloration patterns are classified into groups based on the extent to which changes are observed. For scientific purposes, ten-grade scales are used [4,5], which enables us to capture more natural variability. For use in breeding programs, a four-grade scale is commonly used [14,15]. The highest grade means no yellow marks in abdominal tergites, and the lowest grade marks yellow bands on the first, second, third, and sometimes even on the fourth abdominal tergite. The intermediate grades mean yellow dots on the lateral sides of the second tergite and a yellow band on the second tergite. However, coloration grades obtained as described above are very subjective, and categorical data have lower statistical power.

Morphometric determination of subspecies is labor-intensive since many workers per colony must be evaluated or measured. The preparation of samples is time-consuming, for example, preparation of wings for cubital index measurements, and automatization of some steps in this process is desired to make morphometric subspecies determination more feasible. Automating coloration quantification in *Drosophila*, an important model species, was achieved by using computer-based image analysis [16]. This analysis reduces the error of the subjective grading of the abdominal pigmentation. However, methods for objectively quantifying the abdominal pigmentation in honey bees have not yet been developed. There were some attempts to partially automate the measurement of morphological characteristics of honey bee wings using already developed software for image processing [17], purposely developed software [18], and neural networks [19].

In this work, we developed a computer-based method to objectively quantify abdominal coloration of honey bees on a continuous scale. We tested our method on data obtained from an experiment where we exposed honey bee broods to different environmental temperatures in order to observe phenotypic plasticity in the abdominal coloration of honey bee workers. The developed color-recording protocol and statistical analysis provide useful tools for quantifying abdominal coloration in honey bees.

## 2. Materials and Methods

Using criteria set in the Slovenian Breeding program for *Apis mellifera carnica* [13], we have selected two colonies with the most extreme grades: the colony in which worker bees exhibited yellow marks on abdomens (the “yellow colony”) and the colony without any workers with yellowish patches (the “gray colony”). To obtain the brood of the same age, we enclosed the queens in each colony to empty comb using an isolator cage. We repeated this process twice to obtain two combs of brood from each colony. After capping, the brood was transferred to an incubator. By incubating one comb from each colony at 30 °C and the other at 34 °C, we created four experimental groups: 30gray, 34gray, 30yellow, and 34yellow. The lower temperature was selected through preliminary experiments to determine temperature that still supported hatching most of the brood, and 50 hatched workers from each group were preserved in ethanol and frozen for further analysis.

Abdomens were separated from the body of honey bees and placed in a Petri dish filled with agarose in which we created a small gutter to stabilize the abdomen during imaging. Specimens were arranged in a way that the long axis of the specimen ran in parallel with camera’s x-axis. Images were acquired under a stereo microscope (Leica, Wetzlar, Germany) with constant magnification (8×). Illumination was set to obtain maximum dynamic range of camera to avoid the saturation of pixels. On every abdomen, we manually selected a Region of Interest (ROI) (Figure 1). ROI included entire left half of abdomens to capture as much variability as possible along the length of abdomen. ROIs were chosen using ImageJ software 1.53k [20].

In total, 200 images of honey bee abdomens were generated, 50 for each group. The images were analyzed with a custom-made script using the R programming language [21]. One image was omitted from the analysis because the reflections on the abdomen saturated camera’s sensor. We extracted pixel values for red, green, and blue channels and averaged all three channels to generate a gray-scale image. We normalized pixel values in the range 0–1. Saturated values (values above 0.95) and desaturated values (values that equal 0) were removed. Parts of abdomens with yellow marks had higher pixel values than darker parts of abdomens. Each image was collapsed into a vector of values by calculating median value of each image’s column of pixels as shown in Figure 1. The abdomens varied in length, and the vectors’ lengths were equalized to the longest one, with 1998 values using interpolation function (interp package; [22]).

We performed Uniform Manifold Approximation and Projection (UMAP; [23]) of the vectors to observe any potential structure in this data, separately for each originating colony. We used R package umap [24]. We set the number of components to 2 to graphically present the data and number of neighbors to 75 out of 100 available per group to capture the global variation in the data.

Next, we used Orange software (version 3.27.1; [25]) for a preliminary selection of a supervised classification method of honey bees according to the temperature regime separately for each colony of origin (30 °C or 34 °C). The selected support vector machine (SVM) model with linear kernel was used (R package e1071; [26]). We split the dataset into train and test subsets (75% for training and 25% for testing) to train and evaluate the classifier. We repeated the process of training and testing 10 times for each data input and UMAP parameter. In the first step, unprocessed image vectors were used to build classifier. In the second step, we repeated the process using UMAP preprocessed data to build the classifier. We tested a wide range of UMAP parameter settings. Number of neighbors (5, 10, 20, 50, 75, and 99) balances local and global structure in the data by constraining the number of neighbors to look at. The other variable parameter was the number of UMAP dimensions (from 2 to 10). The number of neighbors was selected to cover the entire range of possible values for both sources of brood (gray and yellow colony). Different numbers of components were selected to extract as much variability in the data as possible. To evaluate the performance of the resulting SVM classifier, we used caret R package [27]. Precision, recall, number of correct predictions across the entire dataset (F1), and area under the ROC curve (AUC) were calculated to evaluate the performance of the classifier.

In the final step, we looked into the possibility of obtaining a single-value measure «coloration index» To obtain coloration index, we calculated the area under the curve when plotting pixel values from vector on y axis and position on abdomen on x axis. We used pracma R package [28] to obtain the area under the curve. To test the normal distribution of coloration indices and residuals, we used Shapiro’s test. F-test was used to confirm the equality of variances for all experimental groups. Since the coloration index data were not normally distributed, we Box–Cox transformed the measurements to obtain a normal distribution. We analyzed variation (ANOVA) in the coloration index between the four groups of incubation temperature and colony origin, including their interaction. Post-hoc Tukey test was performed to compare pairs of experimental groups.

## 3. Results

UMAP was run separately for each colony origin (yellow and gray). Stratification of data was observed between the 30 °C and 34 °C treatments within the gray group (Figure 2A). This stratification was not complete, yet it was indicated that our manipulation of the brood incubation temperature might have affected coloration. We did not observe clear UMAP stratification for the yellow group (Figure 2B).

When we fitted the raw image vector to the SVM classifier, the average performance (Table 1) on workers from the gray colony was the following: precision 0.8 ± 0.00, recall 0.67 ± 0.00, F1 0.73 ± 0.00, and the area under the ROC curve was 0.75 ± 0.00. The performance on workers from the yellow colony was the following: precision 0.69 ± 0.00, recall 0.75 ± 0.00, F1 0.72 ± 0.00, and the area under the ROC curve was 0.71 ± 0.00. To increase the performance of the SVM classifier, we used UMAP-preprocessed image vectors, providing for the gray colony precision 0.80 ± 0.011, recall 0.68 ± 0.053, F1 0.74 ± 0.034, and the area under the ROC curve 0.76 ± 0.026 and for the yellow colony precision 0.91 ± 0.00, recall 0.83 ± 0.00, F1 0.87 ± 0.00, and the area under the ROC curve 0.88 ± 0.00 (Table 1).

The coloration index was conceived as a simple single-value metric for coloration. In the 30gray group, the mean was 441.9 ± 3.4 and 454.0 ± 3.0 in the 30yell group (Table 2 and Figure 3). In the 34gray group, the mean was 446.7 ± 2.7 and 458.1 ± 2.7 in the 34yell group. The ANOVA revealed a significant effect of brood origin on coloration (*p* = 4.03 × 10^−0.5^). The effect of temperature was not significant (*p* = 0.08), nor was the effect of interaction between brood origin and temperature (*p*-value = 0.77). The Tukey test confirmed significant differences among different brood origins. A significant difference was found between the 30gray and 30 yell groups (*p*-value 0.01) and between 34gray and 34yell (*p*-value 0.03). Even though the differences between the two temperatures within the same origin are not statistically significant, minor differences can be observed in Figure 3 and Table 2.

## 4. Discussion

The traditional perception of beekeepers in the *Apis melifera carnica* region is that Carniolan honey bees with yellow abdominal tergites are hybrids with Italian subspecies, and colonies are often excluded from performance testing and breeding just on the basis of coloration. Earlier experiments in other bee species and subspecies, as well as our results, showed that brood nest temperature could be one of the reasons for changes in coloration [7,8], which could make color-exclusion criteria void.

We analyzed image vectors representing coloration along the entire length of workers’ abdomens. Based on our image-embedding method, we could roughly distinguish between the 30 °C groups and 34 °C groups for the gray colony but not for the yellow colony. Similar results were also obtained regarding SVM classification, where we improved the classification performance using UMAP preprocessed data. However, the performance of the SVM classification depended heavily on the combination of UMAP parameters (number of neighbors and number of components). Based on the results of the image vector analysis, we can conclude that the differences between incubation temperature treatments were indicative but not significant.

Since we observed indicative differences in coloration between workers incubated at higher and lower temperatures, we further developed the “coloration index”. Employing the developed coloration index as a measure of coloration did not show the effects of temperature manipulation on the coloration of honey bees’ abdomens. We speculate that the performance of those two methods might have been significant if the temperature difference between the experimental groups was higher, as in Tsuruta et al. [8]. However, the differences in coloration index between the yellow and gray experimental groups were statistically significant. The statistically significant difference in the coloration index values between bees originating from the colony with yellow marks on the abdomen and the one without provides many possible applications. The method, with further validations, has potential for discriminating subspecies.

Our results suggest that origin of the brood has greater influence on coloration of abdominal tergites in honey bees than environment. However, bees from both tested colonies became indicatively lighter in color when incubated at higher temperatures. Change in abdominal color could be assigned to phenotypic plasticity [9]. The influence of genetic background on coloration was also confirmed in experiments conducted by Tsuruta et al. [8]. We also observed that honey bee workers originating from colonies that exhibited yellow phenotypes were lighter in color at higher and lower temperatures. Interestingly, Tsuruta et al. [8] also observed differences in sensitivity to environmental temperature. The great variability between individuals within the experimental groups that originated from the same colony could also be due to different genetic backgrounds of individual workers in the colonies (different patrilines).

Possibly, the main reason for the non-ideal separation between groups is that, in comparison to the study by Degrandi-Hoffman et al. [7], a narrower temperature range was used in our experiment. We incubated the honey bee broods at 30 °C and 34 °C, while Tsuruta et al. [8] incubated the broods at 25 and 38 °C. On the other hand, Degrandi-Hoffman et al. [7] tested an even narrower temperature range, between 31.1 °C and 34.4 °C. Their results were also not statistically significant. Moreover, Tsuruta et al. [8] conducted their experiment on *Apis cerana*, and Degrandi-Hoffman et al. [7] conducted their experiment on *Apis mellifera* queens. Great differences were found in nest temperature in *Apis cerana japonica*: 33.9 ± 0.3 °C in the center of the brood nest, and 31.3 ± 1.7 °C on the periphery of the nest. The differences in the nest temperature in *Apis mellifera* were between 34 °C and 36 °C [29]. Tsuruta et al. [8] reported that this difference in temperature between the periphery and center could be the cause of the variability in color among the adult worker bees. Our decision to use a narrower temperature range was based on preliminary trials, in which we were not able to obtain a sufficient number of workers for further analysis if the temperature was outside the range of 30 °C to 34 °C.

We developed a system to objectively measure the abdominal coloration of the bees on a continuous scale. Continuous data have more information and power from a statistical point of view than categorical data [30]. Measuring abdomen coloration on a continuous scale enables detecting more variability than grading. Vectors obtained from the gray-scale matrix are good input data for training models in machine learning for detecting fine patterns in abdominal coloration of honey bees. Continuous data are also good input data for studies that correlate genomic and phenotypic data (genome-wide association studies, for example). These studies could help us better understand the genetic background of phenotypic traits. In addition, we also eliminated some human errors from the assessment of coloration, thus minimizing the overall errors. However, the developed system is aimed at use in support institutions (e.g., extension organizations and research organizations) since photographing of abdomens requires expensive laboratory equipment. Continuous data for coloration, coupled with genomic information, could be good entry data for genome-wide association studies.

A similar approach towards quantifying abdominal pigmentation in *Drosophila americana* was used by Wittkop et al. [31], who analyzed gray-scale images of 20 regions form flies’ abdomens. This approach was found to be useful for distinguishing among phenotypes. However, their method includes time-consuming sample preparation that includes dissection and mounting of cuticula samples. We adopted sample preparation as described in Saleh Ziabari et al. [16], which is simple and quick since entire honey bee abdomens are simply placed in a Petri dish.

Our results suggest that phenotypic plasticity of abdominal coloration in honey bees could follow Bergmann’s rule [32], even though the differences between the experimental groups are not significant. Phenotypic plasticity of pigmentation and body size due to a change in environmental temperature is summed up in Bergmann’s rule. Bergmann’s rule states that individuals from higher altitudes or latitudes (representing colder environments) are darker and bigger than individuals from hotter environments [32]. The main role of extra melanin in cuticles of poikilothermic animals is to retain more solar radiation and thus elevate body temperature faster [32].

The influence of environmental temperature on abdominal coloration in honey bees could also be observed on a subspecies level. Rosa et al. [33] described lighter color in *Apis mellifera scutellata*, which live in warm and arid climates, in comparison to *Apis mellifera capensis*, which live in the Mediterranean climate with wet winters. Gruber et al. [9] reported similar observations: the morphometric properties of honey bees in Ethiopia vary with altitude. At lower altitudes, honey bees express yellow phenotypes (*Apis mellifera scutellata*), and, at higher altitudes, bees express darker phenotypes (*Apis mellifera monticola*).

Crill et al. [34] described similar phenotypic plasticity in *Drosophila*. Flies incubated at lower temperatures were lighter in color than those incubated at higher temperatures. Furthermore, the temperature to which flies were exposed during development influenced many other physiological and morphological traits. In general, all the changes were in agreement with Bergman’s rule (smaller body size and smaller egg size). Freoa et al. [35] demonstrated that cuticular pigmentation clearly influences body temperature in *Drosophila*. Furthermore, Ref. [36] proved that lower melanization levels in cuticula due to higher environmental temperature are associated with lower melanization of wounds as an initial response of the immune system in insects. Their study was conducted on hemimetabolous grasshopper *Melanopolus sanguipines*. The underlying mechanism of the phenomenon is lower phenoloxidase activity that enhances melanization both in cuticula as well as in terms of an immune response. Circannual changes in day length, humidity, and environmental temperature are reported as causative factors for seasonal morphological variability in lepidopteran species [37].

Aside from coloration, wing venation is another important trait for morphological determination of honey bee subspecies, heavily influenced by the environment and biotic factors. Infestation with ectoparasitic mite *Varroa destructor*, temperature, cell size, and food availability play an important role regarding wing veins layout [38,39,40,41]. The interaction between all the above-mentioned environmental parameters was shown to affect the seasonal variability in wing venation [42,43].

The temperature to which honey bees are exposed as pupae also influences their physiological capabilities. Individuals exposed to lower temperatures (32 °C) as pupae are worse foragers than those exposed to higher temperatures since they complete only 20% of waggle dance circuits in comparison to bees that were raised at higher temperatures [44]. Individuals reared at lower temperatures are more susceptible to intoxication with dimethoate [45]. Environmental temperature during pupation is also important for synaptic organization in the brains of adult honey bees [46]. Nest temperature also influences the rate of bees’ development. Worker pupae that were located at the edge of the brood nest required on average 2 days more to achieve adult stage [47]. Our results are not conclusive yet are important in understanding the effect of environmental temperature on the abdominal coloration of honey bees. Further studies are needed to confirm the influence of environmental temperature on the abdominal coloration of adult honey bees as well as the effects of other environmental factors.

Even though we are deep into the genomic era, quality phenotypic data are still required in evolutionary and physiological studies as well as in animal breeding to improve desired traits. Our method of quantifying abdominal coloration enables us to obtain coloration measurements on a continuous scale and eliminate human factors. Moreover, our method is quick and efficient, enabling us to process a large number of images.

## Figures and Tables

**Figure 1 insects-15-00213-f001:**
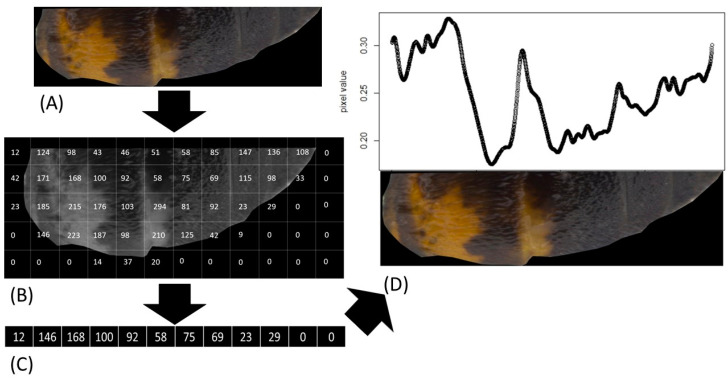
Workflow of our image analysis. Numbers on the image (**B**) are just to demonstrate the concept. Selected ROI is entire half of abdomen (**A**). Then, we extracted matrix of pixel values for all three color channels (red, green, and blue) and averaged them to obtain gray-scale image (**B**). Next step was calculating median across all columns of pixels in the gray matrix and storing the median values in a vector (**C**). Vector of median values represents the coloration on entire length of abdomen. Higher pixel values represent lighter portions of abdomen (**D**).

**Figure 2 insects-15-00213-f002:**
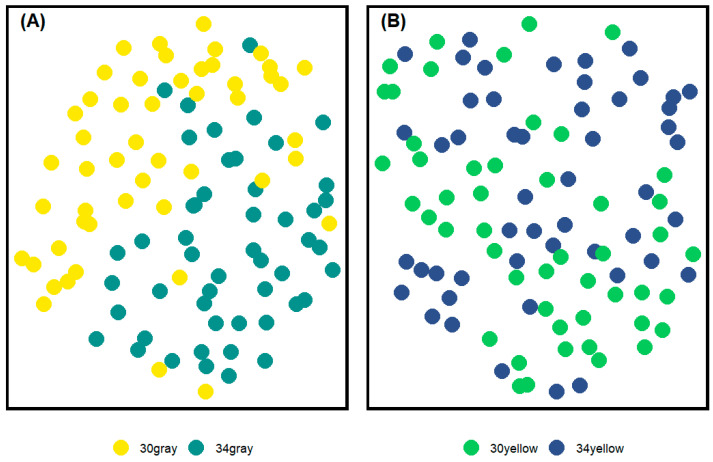
Results of UMAP analysis. Left (**A**) a projection of data obtained on abdomens of workers from the gray colony. Right (**B**) a projection of data obtained on abdomens of workers originating from the yellow colony. Different colors of dots represent the experimental groups.

**Figure 3 insects-15-00213-f003:**
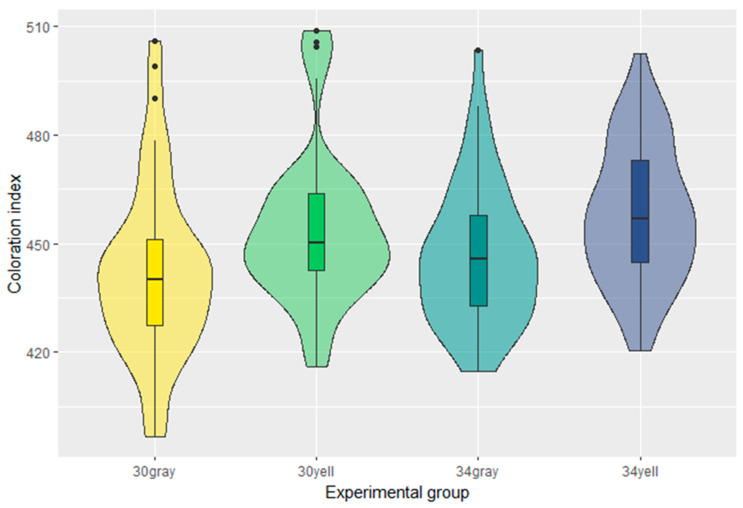
Summary statistics of coloration index by experimental groups. Higher coloration index indicates lighter abdomen.

**Table 1 insects-15-00213-t001:** Performance of SVM model for classification. The model was tested on raw data from gray (gray raw) and yellow colony (yellow raw) and on UMAP preprocessed data from gray (gray UMAP) and yellow colony (yellow UMAP). Training and evaluation of the model were completed in 10 replicates for each dataset.

Group/Metric	Precision	Recall	F1	AUC
Gray raw	0.8 ± 0.00	0.67 ± 0.00	0.73 ± 0.00	0.75 ± 0.00
Yellow raw	0.69 ± 0.00	0.75 ± 0.00	0.72 ± 0.00	0.71 ± 0.00
Gray UMAP	0.80 ± 0.011	0.68 ± 0.053	0.74 ± 0.034	0.76 ± 0.026
Yellow UMAP	0.91 ± 0.00	0.83 ± 0.00	0.87 ± 0.00	0.88 ± 0.00

**Table 2 insects-15-00213-t002:** Summary statistics of coloration index presented by group. N is number of samples; mean represents the mean of experimental group; SD represents standard deviation; Min represents minimum in the experimental group; Max represents maximum in the experimental group; and SE represents standard error.

Group/Metric	N	Mean	SD	Min	Max	SE
34gray	50	446.7	19.3	414.9	503.4	2.7
34yellow	50	458.1	19.2	420.4	502.5	2.7
30gray	50	441.9	23.9	396.7	506.2	3.4
30yellow	49	454.0	21.1	416.0	508.8	3.0

## Data Availability

Code is available at public GitHub repository at the link below. https://github.com/BeeKIS/coloration-assessment accessed on 15 March 2024.

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
