# Peer review of "Quantifying Abdominal Coloration of Worker Honey Bees"

_insects, 2024, doi:10.3390/insects15040213_

Round 1

Reviewer 1 Report

Comments and Suggestions for Authors

The authors evaluating the impact of environmental temperature on honey bee abdominal coloration and developing a tool for quantifying this coloration. The study addresses an important aspect in honey bee subspecies identification, considering the limitations related to genetic background, subjectivity in color grading, and environmental effects.

Here are some comments:

Introduction and methodology

1.     The introduction mentions a lack of knowledge about the genetic background of abdominal coloration. Could you briefly discuss the existing knowledge or hypotheses regarding the genetic basis of abdominal coloration in honey bees?

2.     The connection between environmental factors, particularly temperature, and abdominal coloration is highlighted. It would be beneficial to clarify if this impact is seen during a specific developmental stage or if it persists throughout the bee's life. Additionally, are there other environmental factors beyond temperature that may influence coloration?

3.     Could you provide more details on the criteria for selecting honey bee brood frames from the two colonies? Were there specific characteristics or conditions considered in this selection process?

4.     In the ANOVA analysis, it would be helpful to include details on the sample size for each experimental group. Additionally, were there any statistical tests performed to assess the variability within each experimental group?

5.     Since UMAP did not distinguish individual abdomens according to the experimental group, could you elaborate on potential reasons for this lack of distinction? Were there any challenges or limitations encountered during the UMAP analysis?

6.     The statement that "environmental temperature in the selected range during development does not seem to impact honey bee coloration significantly" is intriguing. Could you discuss potential reasons for this finding, and are there any existing studies that align or contradict these results?

7.     How robust is the developed color recording protocol? Are there specific considerations or potential limitations that researchers should be aware of when applying this tool to different honey bee populations? Could you provide additional information on how the yellow marks on the abdomen were assessed or identified? Were there any potential sources of bias in this process?

8.     Considering the results, are there any plans for further investigations or refinements in the methodology? Are there other environmental factors or genetic aspects that might be considered in future studies?

Results

The results provide valuable insights into the impact of brood incubation temperature on honey bee coloration, and addressing these points could enhance the interpretation and broader implications of the findings.

    1. The UMAP analysis revealed stratification between 30°C and 34°C treatments within the gray group, suggesting an impact of brood incubation temperature on coloration. However, it's noted that the stratification was not complete. Can you discuss potential reasons for the incomplete stratification and whether this might be attributed to biological variability or other factors?
    2. Could you discuss the potential reasons behind the enhanced performance with UMAP preprocessing? Were there specific features in the data that UMAP helped to capture?
    3. The ANOVA results show a significant effect of brood origin on coloration, but not for temperature. Given the trends observed in the coloration index, are there biological or ecological explanations for why temperature differences might not be statistically significant, yet still noticeable?
    4. Are there other environmental factors or genetic aspects that might be considered in future studies?

Discussion

1.     The discussion starts by addressing traditional perceptions of Carniolan honey bees with yellow abdominal tergits being considered hybrids. This contextualization is valuable. Can you discuss how your findings challenge or support these traditional perceptions? Additionally, are there broader implications for breeding programs and colony performance testing based on coloration?

2.     The discussion of UMAP and SVM classifier results is insightful. Could you elaborate on potential biological reasons for the differences in performance between gray and yellow colonies? Are there specific features or patterns in the data that might explain the observed variations in classification performance?

3.     The introduction of the "coloration index" is a significant aspect of your study. The lack of a significant effect of temperature on the coloration index is noted. Can you discuss potential reasons for this discrepancy and speculate on what conditions might yield significant effects, as suggested by Tsuruta et al.? How might the temperature range used in your experiment contribute to this?. 

4.     The discussion links your findings to Bergmann's rule and the concept of phenotypic plasticity. This connection is intriguing. Could you delve into the potential ecological implications of honey bee coloration following Bergmann's rule? How might this influence their adaptation to different environments?

5.     The emphasis on obtaining continuous data for coloration is highlighted. Can you elaborate on the potential applications of continuous coloration data in genomic studies, and how it might contribute to a deeper understanding of the genetic basis of coloration in honey bees?

6.     The comparison with Drosophila studies brings a broader perspective. How might the findings in Drosophila regarding phenotypic plasticity and coloration be relevant or different from those in honey bees, and what insights can be drawn from these comparisons?

7.     The discussion could include suggestions for future research directions. Are there specific aspects that the study did not cover, or additional experiments that could further validate or extend the findings? Address potential confounding factors that might influence coloration, such as diet, genetic factors, or hive conditions. This could strengthen the study's conclusions.

Supplementary files

Supplementary files are not available review (Mentioned in the manuscript but not available to review). the missing supplementary materials (Figure S1, Table S1, Video S1) or clarify the status of these files.

Indicate whether the code used for data analysis, statistical tests, and figure generation is available. If so, specify where readers can access the code, such as through a public repository (e.g., GitHub) or as supplementary material.

Reviewer 2 Report

Comments and Suggestions for Authors

Manuscript title: “Influence of environmental temperature on abdominal coloration of worker honey bees”

Manuscript Number: insects-2880580

Review Report: 

This manuscript is technical in nature and presents a tool to qualify and quantify bee abdominal pigmentation (coloration on the abdominal tergits) in an effort to automate this process and restrict human errors when such evaluation is visually conducted. It is a well-established fact that coloration is mainly linked to the genetic background with potential deviation when strong environmental factors are exerted. Authors acknowledged that in their text and discussion. I do agree that honey bee phenotypic characteristics are very important to track (L333-337) despite the significant advancement in genetic analysis witnessed today. 

The manuscript is clear and acceptably written in general. There are some typos to clear out though. While technically sounds, it does not comprise significant biological findings. Therefore, the title should be changed to reflect this (e.g. Automated tool to measure/differentiate honey bee abdominal pigmentation, or something similar). Since pigmentation and its shift vis-à-vis environmental factors has already been tackled in previous research, why authors diluted the significance of their Imaging tools to re-address a biological question? This is not clear to me and is a major drawback. The finding of this work is no that “Our results indicate that environmental temperature in selected range during development does not seem to impact honey bee coloration significantly”, this is already established. Your focus should be on the tool you are presenting “The developed color recording protocol and statistical analysis provide a useful tool for quantifying abdominal coloration in honey bees”. Such color protocol proposed here should be the core of your study and should have been tested against different honey bee subspecies samples, showing already wide range of color variation, which will enable a more robust evaluation and vetting of your proposed tool. 

Major Concerns: 

1-    I am ready to endorse this study, and I believe it is relevant, but the whole biological aspect of it (comparing pupations phases at 30C vs 34C) does not add anything meaningful. As you saw, most values are not significant too. You can keep the experimental design as it is, but the whole manuscript and its discussion must be shifted and refocused around the “color recording protocol” you are proposing. 

2-    Discussion should be reoriented to compare your tool with previously use/developed tool for measuring pigmentation (e.g. compare yours with Ruttner’s1987 or others).

3-    The best remedy for this study is to include and analyze samples from other subspecies as mentioned above.      

Minor issues:

1-    Italicizes all Latin names across the text. None is. 

2-    UMAP acronym was never developed in the text, please develop at first use (L12) “Uniform Manifold Approximation and Projection”? 

3-    L324: “We then calculated are under”, correct, something is missing.

4-    L274: delete the dot after mellifera. 

5-    L292: varroa mite does not fall under “environmental stressor”, it is a biotic stressor. Please consider.  

Comments on the Quality of English Language

Minor edits and improvement required.  

Reviewer 3 Report

Comments and Suggestions for Authors

Brief:

            Bubnič and Prešern present an original and interesting study aimed to evaluate the effect of environmental temperature on the abdominal coloration of honeybee workers and to develop a tool for quantifying abdominal coloration. Although the differences in mean coloration index were not significant between bee groups incubated at different temperatures, authors introduced a color recording protocol and statistical analysis that was tested as an interesting and valuable tool for quantifying abdominal coloration in honey bees. The article is undoubtedly worth to be published, so I recommend it for publication. Only some minor corrections are needed (see bellow).

Specific comments:

Introduction

Line 47: Insert ¨about¨ or ¨regarding¨ between ¨ …. knowledge¨ and ¨its…..¨.

Materials and Methods

Line 116: Replace ¨such¨ with ¨in a way¨

Line 120: Here in the text, you define the Region of Interest (ROI) and refer it to Figure 1. However, the ROI indication is missing on the Figure 1 or in Figure 1 text. Please, mark ROI on the Figure 1 or remove ROI from the text.

Line 128: delete ¨were¨

Line 136: Unify the way the ¨umap¨ is written in the text. Here, it should be converted to the capital letters.

Results:

Figure 1. D: Pixel values on vertical scale are missing. Please, add.

Figure 1 text: Line 170: I don’t see any numbers in image A. Please, correct image or text.

Table 1, 1st row: The last column name is missing, please name/define the values 2.7, 2.7 etc.

Discussion:

Lines 251-252: insert between ¨….in comparison to¨ and ¨ [4,5] ¨ the names of the authors of citations numbered 4 and 5.

Lines 266: Nevertheless, the reference regarding Bergman’s rule is included later in the text, it should be included also here, where this rule is first mentioned.

Round 2

Reviewer 1 Report

Comments and Suggestions for Authors

The authors satisfactorily addressed my concerns, and I do not have any further comments.

Author Response

Dear Reviewer,

thank you again for valuble comments that improved our manuscript.

Kind regards, 

Jernej Bubnič